# Early N-Terminal Pro B-Type Natriuretic Peptide (NTproBNP) Plasma Values and Associations with Patent Ductus Arteriosus Closure and Treatment—An Echocardiography Study of Extremely Preterm Infants

**DOI:** 10.3390/jcm11030667

**Published:** 2022-01-27

**Authors:** Anna Gudmundsdottir, Marco Bartocci, Oda Picard, Joanna Ekström, Alexander Chakhunashvili, Kajsa Bohlin, Caroline Attner, Gordana Printz, Mathias Karlsson, Lilly-Ann Mohlkert, Jonna Karlén, Cecilia Pegelow Halvorsen, Anna-Karin Edstedt Bonamy

**Affiliations:** 1Department of Women’s and Children’s Health, Karolinska Institutet, 171 77 Stockholm, Sweden; marco.bartocci@ki.se (M.B.); oda.blomquist-picard@regionstockholm.se (O.P.); gordana.printz@gmail.com (G.P.); 2Department of Neonatology, Astrid Lindgren Children’s Hospital, Karolinska University Hospital, 171 64 Stockholm, Sweden; kajsa.bohlin@ki.se; 3Emergency Paediatric Emergency Medicine, Children’s Minnesota, Minneapolis, MN 55404, USA; joanna.ekstrom@gmail.com; 4Department of IT Development and Healthcare Information Systems, Karolinska University Hospital, 171 64 Stockholm, Sweden; alexandre.chakhunashvili@regionstockholm.se; 5Department of Clinical Science, Intervention and Technology, Karolinska Institutet, 171 77 Stockholm, Sweden; lilly-ann.mohlkert@regionstockholm.se; 6Sachs’ Children and Youth Hospital, Södersjukhuset, 118 83 Stockholm, Sweden; caroline.attner@regionstockholm.se (C.A.); jonna.karlen@ki.se (J.K.); cecilia.pegelow-halvorsen@regionstockholm.se (C.P.H.); 7Clinical Research Center and Department of Clinical Chemistry, Central Hospital, 652 30 Karlstad, Sweden; mathias.karlsson@equalis.se; 8Department of Clinical Science and Education, Karolinska Institutet, Södersjukhuset, 118 83 Stockholm, Sweden; 9Clinical Epidemiology Division, Department of Medicine Solna, Karolinska Institutet, 171 77 Stockholm, Sweden; anna-karin.edstedt.bonamy@ki.se

**Keywords:** cardiac troponin T, echocardiography, extremely preterm, N-terminal pro B-type natriuretic peptide, patent ductus arteriosus

## Abstract

The aim was to investigate the association of gestational age (GA), echocardiographic markers and levels of plasma N-terminal pro-B-type natriuretic peptide (NTproBNP) with the closure rate of a haemodynamically significant patent ductus arteriosus (hsPDA). Ninety-eight Swedish extremely preterm infants, mean GA 25.7 weeks (standard deviation 1.3), born in 2012–2014, were assessed with echocardiography and for levels of NTproBNP. Thirty-three (34%) infants had spontaneous ductal closure within three weeks of age. Infants having spontaneous closure at seven days or less had significantly lower NTproBNP levels on day three, median 1810 ng/L (IQR 1760–6000 ng/L) compared with: infants closing spontaneously later, 10,900 ng/L (6120–19,200 ng/L); infants treated either with ibuprofen only, 14,600 ng/L (7740–28,100 ng/L); or surgery, 32,300 ng/L (29,100–35,000 ng/L). Infants receiving PDA surgery later had significantly higher NTproBNP values on day three than other infants. Day three NTproBNP cut-off values of 15,001–18,000 ng/L, predicted later PDA surgery, with an area under the curve in ROC analysis of 0.69 (0.54–0.83). In conclusion, the spontaneous PDA closure rate is relatively high in extremely preterm infants. Early NTproBNP levels can be used with GA in the management decisions of hsPDA.

## 1. Introduction

In extremely preterm infants, diagnostic and treatment strategies of a haemodynamically significant patent ductus arteriosus (hsPDA) have changed, with fewer infants treated and at a later postnatal age [1]. The definition of a hsPDA has been based on echocardiographic signs of pulmonary overload along with end-organ steal, as well as clinical signs such as heart failure and circulatory instability [2,3]. The definition of a hsPDA has varied between studies and many studies lacked longitudinal echocardiographic data and thereby knowledge about the duration of the PDA shunt [4,5]. The evidence for different PDA treatment strategies is weak and data on the association with later outcomes is limited [6]. However, centers without any intervention to promote ductal closure observed a high spontaneous ductal closure rate (73–97%) in infants born before 28 weeks [7,8,9]. The predictive value of clinical and echocardiographic markers of ductal severity has been studied in association with adverse outcome, and the additive role of the cardiac biomarkers suggested a possible importance of N-terminal pro B-type natriuretic peptide (NTproBNP) [10,11,12,13]. Another biomarker, cardiac Troponin T (cTnT), has been suggested to be relevant, but studies have been inconclusive on the association with hsPDA [14]. Both NTproBNP and cTnT have been associated with myocardial function in the preterm population [15], and increased NTproBNP and cTnT values during postnatal adaptation in newborns have been shown [16,17]. NTproBNP is associated with expansion of the cardiac walls in infants with volume and pressure overload [18] and it has been suggested that the shunt through the PDA with overload on left heart chambers has similar effects, with elevated levels of NTproBNP [13]. Relatively low levels of NTproBNP have been reported to be associated with early spontaneous ductal closure in very preterm infants but studies on extremely preterm infants are sparse [19,20].

We hypothesised that early in life levels of NTproBNP and cTnT predicted which extremely preterm infants would either be exposed to pharmacological PDA treatment or close their duct spontaneously. To study this, we investigated a cohort of infants born from 23 + 0 to 27 + 6 weeks of gestation with consecutive echocardiography assessments and measurements of NTproBNP and cTnT.

## 2. Materials and Methods

### 2.1. Participants

This is a prospective hospital-based cohort study of extremely preterm infants, born before 28 weeks of gestation and admitted to the neonatal intensive care unit at Karolinska University Hospital in Stockholm, Sweden. Infants born between 14 February 2012 and 31 July 2014 were consecutively evaluated for inclusion and of 169 eligible infants, 105 were included in the study. Infants who had cardiac or other major malformations were excluded. Only infants with NTproBNP and cTnT measurements available (*n* = 98) were included in the analyses, Figure 1.

### 2.2. Echocardiographic Assessment of the PDA 

All eligible infants had a comprehensive echocardiographic assessment to rule out structural heart defects. Echocardiography was performed, based on suggested standardised views for pediatric echocardiograms [21,22] on day three, day seven, day 14 and at regular intervals according to a predefined clinical protocol, which was a simplified version of the scoring system by McNamara et al. [3], Appendix A, until the ductus was closed. Furthermore, infants treated for hsPDA were also examined before and after treatment. If echocardiography was not performed at the predefined timepoint or if several assessments were performed, an age-range within 2–4 days and 5–9 days was accepted as day three and day seven, respectively. Echocardiographies were performed by neonatologists specially trained and, in addition, infants needing PDA surgery had a complete pre-surgery echocardiography performed by a specialist in paediatric cardiology. All cardiac assessments were performed with a 10V4 and 8V3 MHz transducer, using Siemens Acuson, S1000 or SC2000 systems (Siemens Medical Solutions, Ann Arbor, MI, USA). Syngo Dynamics Workplace (Siemens Medical Solutions, USA) was used for off-line image analyses. The study investigator with extensive training in echocardiography (AG) re-examined all the echocardiographies according to a predefined protocol (designed by L-AM and AG) and AG was blinded to the biomarker results. These were the measurements accepted for the final analyses. The echocardiographic variables described in Appendix A, were as follows: ductal diameter, left atrial to aortic root ratio (LA:Ao ratio) and maximal velocity (Vmax) through the duct. The end-diastolic flow in the right or left pulmonary artery was dichotomised as more than 0.2 m/s or 0.2 m/s or less. Reversed diastolic flow in the descending thoracic aorta or reversed or absent diastolic flow of the mesenteric artery or anterior cerebral artery, were dichotomised as any sign of end-organ steal versus none. 

### 2.3. Shunt Duration with or without PDA Treatment

The exposure was the presence and shunt duration of an echocardiographically verified PDA, either haemodynamically significant or not. The outcomes studied were age in days at PDA closure and PDA treatment, classified as pharmacological PDA treatment with ibuprofen only, or all PDA surgery. Closure of the PDA was determined as no ductal flow detected with colour Doppler on echocardiography. The date of PDA closure was defined as the date of the examination confirming PDA closure, or date of PDA surgery. Infants having no confirmed PDA closure with echocardiography and no new examination after 36 weeks postmenstrual age, were assigned a closure age at full-term 40 weeks.

Decisions on any hsPDA treatment, pharmacological, surgical, or both, were made by the neonatologist in charge, using a combination of echocardiography data and clinical criteria, as recommended by the local regional hospital guidelines for PDA pharmacological treatment, Appendix A. For pharmacological treatment, ibuprofen (Pedea^®^ 5 mg/mL, Orphan Europe Nordic, Stockholm, Sweden) was used and administered intravenously in doses of 10 mg/kg on the first day of treatment, followed by 5 mg/kg the next two days. Echocardiography was performed after a course of three doses and a decision was made if further treatment (another three doses) was indicated according to the clinical regional guidelines.

### 2.4. Cardiac Biomarkers

The cardiac biomarkers NTproBNP and cTnT were measured on day three and seven. If several measurements were available or the blood sampling had been deferred to synchronise with a clinically indicated test, the same approach was applied as for the echocardiography time-points, as described above. The first ten included infants had blinded biomarker measurements but due to blood loss concerns a protocol change was made with biomarkers measured with other clinically indicated blood samples. As the biomarkers were automatically registered in the electronic medical chart, the neonatologist in charge was not blinded to the results.

Blood samples for cTnT and NTproBNP were collected in 0.5 mL lithium heparin tubes and analysed at the accredited laboratory at Karolinska University Hospital. The analyses of NTproBNP were performed using a highly sensitive NTproBNP assay (Electrochemoluminience Cobas e602, Roche Diagnostics; https://diagnostics.roche.com/global/en/products/params/elecsys-nt-probnp.html, accessed on 26 January 2022), measured in ng/L with minimum and maximum limits of 5 ng/L and 35,000 ng/L respectively. The analyses of cTnT were performed using a highly sensitive cTnT assay (Electrochemoluminience, Cobas e602, Roche Diagnostics; https://diagnostics.roche.com/global/en/products/params/elecsys-troponin-t-high-sensitive-tnt-hs.html, accessed on 26 January 2022) and measured in ng/L, with a minimum detection limit of 5 ng/L.

### 2.5. Infant Characteristics

Data on gestational age, birth weight and birth weight standard deviation score (SDS) according to the Swedish reference curve for normal intrauterine growth [23], antenatal corticosteroid treatment, multiple birth, mode of delivery (vaginal or cesarean), any surfactant administered within 2 h after birth, Apgar score at five minutes of age and presence of chorioamnionitis as confirmed by histopathological examination, were collected. All data on perinatal and neonatal characteristics, as well as results of the biomarker measurements, were extracted from infants’ electronic medical charts.

The study was conducted in accordance with the Declaration of Helsinki and the protocol was approved by the Regional Ethical Review Board in Stockholm, reference number: 2011/772-31/4, with an amendment 2012/1253-32. A written signed parental consent was obtained before participation.

### 2.6. Statistical Analyses

Results are reported in numbers with proportions (%), means with standard deviations (SD) and medians with interquartile ranges (IQR). The Shapiro–Wilk and Anderson–Darling tests were used to test for normal distribution. To test for differences between groups, Student’s t-test was used for normally distributed data and Wilcoxon rank-sum test, Kruskal–Wallis test and Mood Median for non-normal data. Chi-square test or Fischer’s Exact Probability test was used to examine the differences in proportions. For pairwise post-hoc comparisons, Dunn’s test was used. Kaplan-Meier survival curves were used to describe the incidence of ductal closure over time, spontaneous or after ibuprofen treatment. It was specified by gestational age (GA) and number of ibuprofen treatment doses. The observation time was from birth to 40 weeks postmenstrual age (maximal observation time 133 days). Infants were censored at death, transfer to any hospital in another Swedish region or surgical closure of the PDA, whichever occurred first. Ninety-five percent confidence bands (CB) (STATA command stcband) and 25% and 75% closure rates, when available from survival analysis, were used to describe the variance in ductal closure. The 25% and 75% closure rates were sometimes not available due to the limited number of infants in subgroups and censoring at that timepoint. To compare Kaplan-Meier curves, the Chi-square method with Log-rank and Wilcoxon non-parametric tests was performed. Receiver operating characteristic (ROC) curves were constructed and area under the curve (AUC), with 95% confidence intervals (CI), calculated to estimate the cut-off points of the predictor variables for the outcomes in question. The differences between the ROC curves were tested and the 95% CI compared between the predictors with non-parametric tests (roccomp in STATA). Finally, a correlation analysis was performed to study the relationship between variables, and the Pearson correlation coefficient was calculated. A *p*-value below 0.05 was considered statistically significant.

All statistical analyses were performed using STATA MP 16 (http://www.stata.com/, accessed on 26 January 2022), Microsoft Excel version1808 (https://www.microsoft.com/en-us/microsoft-365/excel, accessed on 26 January 2022) and Minitab Statistical Software Version 17 (http://www.minitab.com, accessed on 26 January 2022).

## 3. Results

In the cohort, the mean gestational age was 25.7 weeks (SD 1.3). Of 58 infants (59%) treated for a hsPDA, 36 (62%) were treated with ibuprofen only, 17 (29%) with ibuprofen followed by later PDA surgery and five (9%) with primary PDA surgery, Table 1. Median age at the start of any PDA treatment was eight days (IQR 5–11). Of the fifty-three infants treated with ibuprofen, 28 (53%) were treated at seven days of age or less, median five days (IQR 5–6), and 25 (47%) were treated at more than seven days of age, median 11 days (IQR 10–12).

Twelve (12%) infants died, at a median age of 14 days (IQR 13–16) with a mean GA of 24.5 weeks (SD 1.3) and a mean birth weight of 632 g (SD 179). Infants not having PDA treatment who later died had lower Apgar scores and had received surfactant directly after birth to a higher extent compared with the other untreated infants, Appendix A.

### 3.1. Ductal Closure 

Seventy-nine (81%) infants had a registered ductal closure date, which was ultrasound confirmed or due to PDA surgery in 76 (78%) of the infants. Three infants were assigned a closure date at full term (40 weeks). Eleven (11%) infants died before confirmed ductal closure and five (6%) were transferred to another Swedish hospital before confirmed ductal closure, Appendix A.

In the whole cohort (*N* = 98), fifty percent had a closed PDA by day 45 (25% closure rate; 75% 20;83) irrespective of PDA treatment. Fifty percent of the infants had a closed PDA by day 42 (17;83) if born at 25 weeks or later; and by day 51 (38;57) if born before 25 weeks but this difference was not significant, Figure 2a. Infants spontaneously closing their duct were all born at 25 weeks of gestation or later.

After excluding the five infants having primary PDA surgery, the remaining 93 infants were dichotomised as receiving ibuprofen treatment or not, Appendix A. There was a significant difference in the 50% closure rate of infants receiving ibuprofen treatment compared to infants not treated, by day 62 (38;115) and by day 24 (13;58) respectively. Furthermore, the infants were categorised into (i) no PDA treatment, (ii) treated with three doses or less of ibuprofen or (iii) treated with more than three doses of ibuprofen, Appendix A. The 50% cumulative probability of ductal closure was reached by day 24 (13;58) for infants closing their PDA spontaneously, by day 55 (15;missing) for infants treated with three doses or less of ibuprofen (*n* = 12, whereof one infant died) and by day 63 (45;115) for infants treated with more than three doses of ibuprofen (*n* = 24, whereof four died), Appendix A. Significant differences were found in overall comparisons of the three curves. Furthermore, in pairwise comparison a significant difference was detected in the group of infants not treated versus infants treated with more than three doses of ibuprofen. Of the seventeen infants who were operated after ibuprofen treatment, twelve (71%) received more than three doses of ibuprofen.

Furthermore, when infants who were transferred, underwent PDA surgery or died, were excluded, the 50% rate of PDA closure was studied in the remaining 60 infants. It was reached by day 21 (11;46) for infants spontaneously closing their duct and significantly differed compared with infants treated with ibuprofen only who had a 50% PDA closure rate by day 51 (17;106); Figure 2b.

Besides studying the 50% closure rate with Kaplan-Meier survival analysis curves, the median age at closure was studied in the different treatment categories, Appendix A.

Lastly, the current GA at ductal closure was studied in this group of surviving infants but infants closing the duct at more than 133 days of age were excluded (*N* = 6). The groups did not differ significantly, the median current GA at ductal closure was 29 weeks (IQR 27–30) in the no treatment group and 32 weeks (IQR 27–36) in the ibuprofen only group.

### 3.2. Echocardiographic Markers of Ductal Severity

Nearly all infants treated either with ibuprofen or surgery met all the predefined criteria of an echocardiographic hsPDA, Appendix A, already on day three, Table 2. Infants who were not treated but later died, had echocardiographic characteristics on day three similar with the characteristics of the group of infants later treated for PDA. Overall comparisons of the echocardiographic markers between the no treatment group and the treatment groups on day three detected significant differences in ductal diameter, LA:Ao ratio, signs of end-organ steal and excessive pulmonary circulation (end-diastolic pulmonary flow > 0.2m/s), as indicated in Table 2, where pairwise comparison between groups is also shown. On day seven, there were significant differences detected in LA:Ao ratio, excessive pulmonary flow circulation and in maximal ductal flow velocity between the no treatment group and the treatment groups in overall comparison (Table 2).

### 3.3. Cardiac Biomarkers

#### NTproBNP

Among the six infants closing their duct spontaneously at day seven or earlier, NTproBNP was significantly lower on day three compared to the value of infants of all the PDA treatment categories, Table 3. Furthermore, infants having PDA surgery later, had on day three significantly higher levels of NTproBNP compared to all other categories at that time-point, Table 3. There were significant differences in both overall and pairwise comparisons between the median values of all infants closing the duct spontaneously at seven days or less and the other treatment categories. In infants spontaneously closing at 7 days of age or earlier, the median NTproBNP on day three was 1810 ng/L (IQR, 1760–6000 ng/L) compared to infants closing spontaneously at more than seven days, 10,900 ng/L (6120–19,200 ng/L); or infants treated for PDA either with ibuprofen only, 14,600 ng/L (7740–28,100 ng/L); or undergoing PDA surgery, 32,300 ng/L (29,100–35,000 ng/L), Table 3. The same was found for infants having PDA surgery when compared with all the other categories, Table 3. There was one exception in the pairwise comparison of no PDA treatment (with open duct beyond first week of life) and ibuprofen treatment only where no significant differences were found in the levels of NTproBNP.

In a separate analysis, levels of NTproBNP in infants having later PDA surgery after prior ibuprofen treatment were compared with infants treated with ibuprofen only and they had significantly higher levels of NTproBNP on day three, 33,500 ng/L (29,500–35,000 ng/L) vs. 14,600 ng/L (7740–28,100 ng/L), respectively *p* < 0.01.

Infants closing their PDA at seven days or less of age had significantly lower levels of NTproBNP on day seven, compared with the other groups. Infants having PDA surgery later had significantly higher levels of NTproBNP compared with infants not treated (and later closing their duct at any timepoint), Table 3. However, on day seven the levels of NTproBNP in the PDA surgery group did not differ significantly from those of infants treated with ibuprofen.

For the ROC analysis of NTproBNP on day three, 80/98 infants had a value available for analysis. When analysed for spontaneous ductal closure the AUC value was 0.31 (95% CI 0.19–0.43), with a sensitivity and specificity 61% and 20%, respectively, for the cut-off value at 6001–9000 ng/L, Figure 3a.

In predicting PDA surgery, a cut-off value of NTproBNP at 15,001–18,000 ng/L resulted in an AUC of 0.69 (95% CI 0.54–0.83), a sensitivity of 66%, and specificity of 66%, Figure 3b.

Furthermore, it was studied how GA at birth, diameter of the duct and LA:Ao ratio on day three predicted spontaneous closure and later surgery. For GA, eighty infants were included in the analysis; the AUC was 0.76 (0.67–0.86) with a sensitivity and specificity of 74% and 63%, respectively, at a cut-off GA at 26 weeks, in predicting spontaneous ductal closure. However, the AUC for predicting all PDA surgery was 0.19 (0.10–0.28), with sensitivity and specificity of 11% and 37%, respectively, with a GA cut-off at 26 weeks. The AUC for ductal diameter in predicting spontaneous ductal closure was 0.18 (0.04–0.33), *N* = 50. For LA:Ao ratio, the AUC was 0.20 (0.06–0.34), *N* = 48. When studying PDA surgery, the AUC was 0.54 (0.35–0.73) for ductal diameter and for LA:Ao ratio 0.69 (0.52–0.86). This is shown in the Appendix A (spontaneous closure) and the figure of the graphical abstract (PDA surgery) of the article. Furthermore, a comparison of the confidence intervals for the AUCs of the different predictors on day three are shown in Appendix A. Lastly, when the correlation of NTproBNP and LA:Ao ratio was studied the correlation coefficient was 0.76 (R^2^ = 0.579), Appendix A.

### 3.4. Troponin T 

No statistically significant differences were found in comparisons of cTnT levels between the different PDA treatment groups, Table 4. The cTnT levels on day three in infants who later died did not differ from infants who survived, 215 ng/L (166–254 ng/L) vs. 159 ng/L (109–261 ng/L), respectively, (*p* = 0.12) but on day seven cTnT levels were significantly higher, 144 ng/L (100–213 ng/L) vs. 96.5 ng/L (83–139 ng/L) with a *p*-value of 0.03.

## 4. Discussion

In this observational cohort study of extremely preterm infants, the proportion of an echocardiographically verified ductal closure was high, both spontaneous and after ibuprofen treatment, although closure sometimes did not occur until near-term age. The biomarker NTproBNP, measured on day three of life, predicted PDA surgery. The importance of gestational age for spontaneous ductal closure observed in previous studies was confirmed [7,8,9]. NTproBNP could be a complement to the gold standard echocardiography, in evaluation of a hsPDA in extremely preterm infants.

Our study reported a high spontaneous closure rate in infants born at 25 gestational weeks or later, consistent with recent reports [8,26]. Due to high treatment rates, we were not able to report the actual spontaneous rate in infants born before 25 gestational weeks in our cohort, but it has been reported to be relatively high in centers with a non-interventional PDA strategy [8,9,26,27,28]. Furthermore, there was possible overtreatment as infants having three ibuprofen doses or less had a similar closure rate as infants who closed their duct spontaneously. This could be of importance when designing future randomized PDA treatment studies as well as when deciding which patients are eligible for percutaneous catheter ductal closure.

In the era of PDA management changes, we could regard the infants in this study that later had PDA surgery, as representatives of infants with the grade of ductal severity that warrants special attention. The use of biomarkers as a complement to echocardiographic scoring systems has been suggested as an attractive approach, as they are neither dependent on echocardiographic resources, nor associated with inter-examiner variability [10]. Studies on the correlation of NTproBNP and cTnT values with echocardiographic signs of hsPDA have been inconclusive [15,29]. In our cohort of extremely preterm infants, we found that NTproBNP on day three predicted later PDA surgery. Furthermore, infants with low levels of NTproBNP on day three closed their ducts spontaneously within seven days of age but this finding needs confirmation in a larger cohort. Optimally, a ROC curve for a diagnostic marker will need better sensitivity and specificity than our results showed for NTproBNP, but the value on day three between 15,001–18,000 ng/L, could serve as a reasonable indication for selecting infants for evaluation of the duct with echocardiography. This is in line with some earlier studies in cohorts of more mature infants, where NTproBNP helped differentiation between infants treated or not for PDA [19,20]. However, this needs to be interpreted cautiously as the positive and negative predictive values of the diagnostic tests are dependent on incidence and, as described earlier, the PDA treatment rate has decreased during the last decade in our and other centers [1]. ROC analyses of GA were limited due to few infants born before 25 gestational weeks and they were all either treated or later died, but many other studies have shown that GA is a strong predictor of spontaneous closure. The usual echocardiographic markers of a hsPDA in our study poorly predicted both spontaneous closure and later surgery which indicates that the echocardiographic evaluation is complex. As the methods for NTproBNP differ between laboratories, it has hitherto been difficult to compare previous and current studies of infants with hsPDA [10]. NTproBNP levels, on day three, could be helpful in selecting which infants warrant further attention for the PDA and we suggest that in future studies the predictive value of NTproBNP in combination with GA and some echocardiographic markers of diastolic dysfunction are examined. Contrary to many neonatal units, our unit did not use functional evaluations to evaluate diastolic and systolic dysfunction in the PDA treatment process, such as tissue Doppler or assessments of cardiac output. The diagnosis of a hsPDA is becoming more complex and would therefore most likely benefit from having advanced diagnostics performed by experienced echocardiographers [30]. The values of cTnT in the infants were high compared to normal values in adults, which has been observed in other studies of newborn infants, and cTnT was neither associated with PDA treatment, nor spontaneous closure of the PDA in our study [16].

A verified hsPDA during the first days of life has been associated with later death and neonatal morbidities such as intraventricular and pulmonary haemorrhage, necrotising enterocolitis (NEC) and bronchopulmonary dysplasia (BPD) [6,31,32,33,34]. The various PDA treatment strategies practiced by clinicians or tested in numerous studies have been driven by the attempts to reduce the risk of ductus-related complications and morbidities [5]. Yet, the evidence for PDA treatment strategies and associated outcomes is insufficient and confounded by unknown rates of early spontaneous ductal closure, early use of rescue treatments and a failure to consider the duration and severity of PDA shunts in previous studies [35].

This study had several strengths. First, in this clinical contemporary cohort of infants, born before 28 gestational weeks, the most complex hsPDA were represented and, therefore, the external validity of the study was strengthened. Second, the longitudinal echocardiographic verifications enabled us to measure shunt time and verify age at ductal closure. By using the Kaplan-Meier method, infants that died before ductal closure were accounted for when estimating the ductal shunt duration. Furthermore, the late start of treatment in the cohort made it possible to study the spontaneous closure rate during the first week in extremely preterm infants. Limitations of the study include that the study protocol allowed inclusion up to 72 h of age. This introduced a bias, as some of the sickest infants that died during the first days of life could not be included. Due to the observational design of the study, the PDA treatment was at the discretion of the neonatologist in charge, although local guidelines were followed using both clinical and echocardiographic criteria (Appendix A). This limited interpretation of the PDA surgery outcome. Moreover, there was heterogeneity in the biomarker data due to the difference in timing of sampling, arising from the need to combine it with clinically indicated blood sampling to address blood loss concerns. For the same reason, the clinicians were not blinded to the biomarker values, which may have influenced PDA treatment decisions. Echocardiography is an operator-dependent assessment and there were in total ten neonatologists performing the echocardiographies of this cohort, which was a limitation. However, all the exams were reviewed off-line by the same neonatologist trained in echocardiography, who was blinded to the biomarker results and the reviewed results were the measurements used for the final analyses. Furthermore, due to the size of this cohort study the power was limited, especially in subgroup analyses of the biomarkers.

Long-term cardiac outcome is of interest, since a more conservative approach to PDA management can be expected to result in a larger group of infants being exposed to ventricular dilatation due to PDA shunts for longer periods of time. Clinical case reports have suggested that an open duct may contribute to pulmonary hypertension associated with BPD in the most immature population with the sickest lungs and register studies have showed an association between extremely preterm birth and childhood risk of heart failure [36,37]. To contribute with long-term data, there is an on-going echocardiographic cardiovascular follow-up of this cohort.

## 5. Conclusions 

Extremely preterm infants have a high ductal closure rate, both spontaneous and after ibuprofen treatment, but closure can be delayed until full-term age. NTproBNP measured during the first postnatal days is helpful in identifying infants at risk of developing a haemodynamically significant PDA, in consideration of treatment. Further studies on PDA diagnostic scores, including clinical signs, echocardiographic markers as well as biomarkers and studying the associations with neonatal morbidities, are needed.

## Figures and Tables

**Figure 1 jcm-11-00667-f001:**
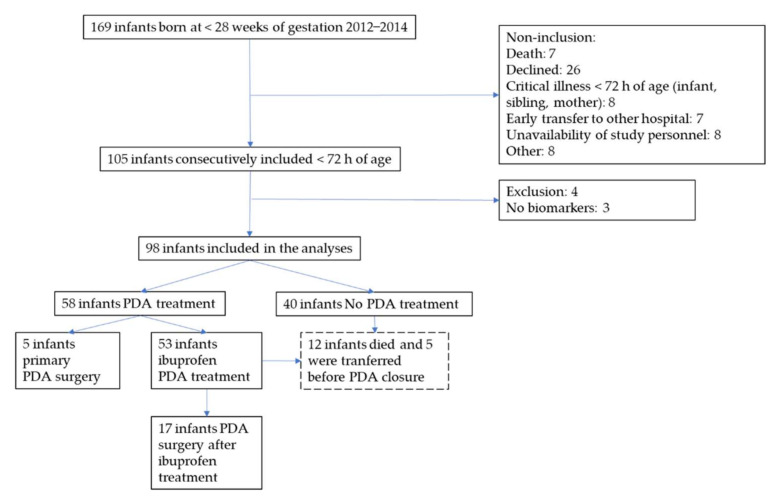
Flow chart of the study showing inclusion and exclusion.

**Figure 2 jcm-11-00667-f002:**
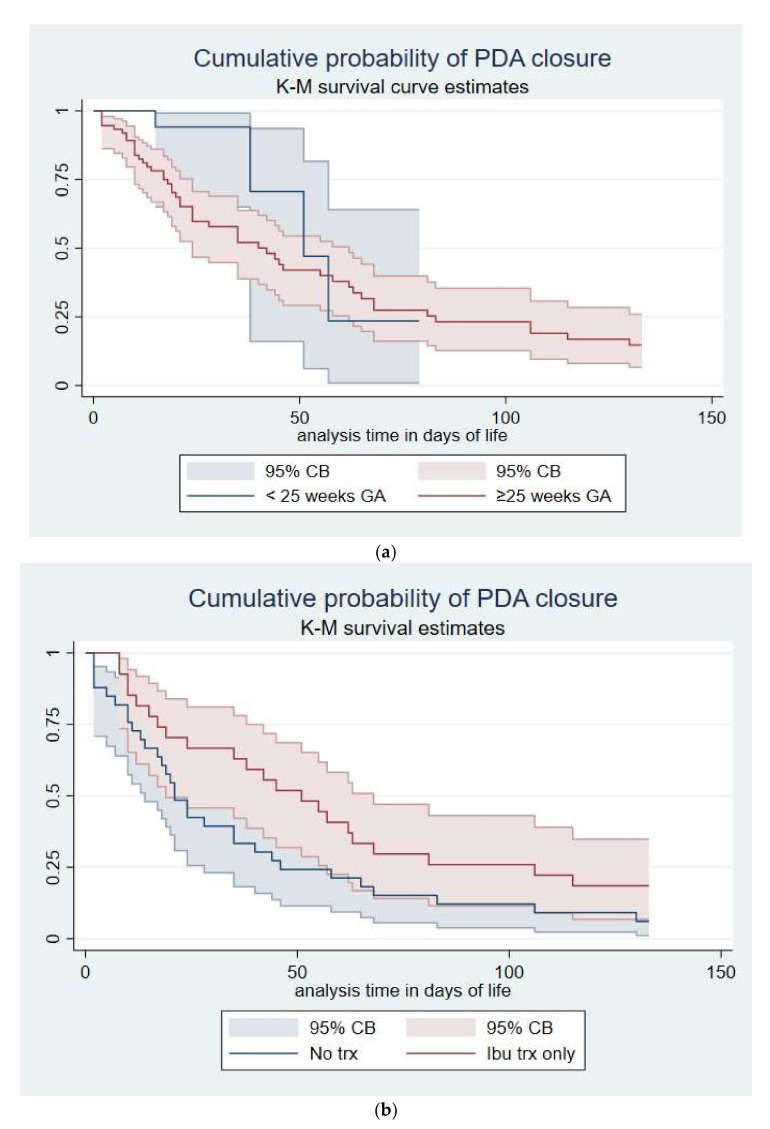
(**a**) The cumulative probability of PDA closure for all the infants (*N* = 98) is shown with a K-M survival curve with 95% CB. Infants are divided into those born before 25 weeks of GA and those born at or later than 25 weeks GA. The analysis time was until 133 days of age as then all the infants had reached 40 weeks postmenstrual age. The censoring protocol is shown in Appendix A. (**b**) The cumulative probability of PDA closure for infants who either were not treated or only treated with ibuprofen is shown in a K-M survival curve with 95% CB. Infants that later died, were transferred or underwent PDA surgery were excluded (*N* = 60). The analysis time was until 133 days of age as then all the infants had reached 40 weeks postmenstrual age. The censoring protocol is shown in Appendix A. Abbreviations: PDA: patent ductus arteriosus; K-M: Kaplan-Meier; trx: treatment; Ibu: ibuprofen, CB: confidence bands.

**Figure 3 jcm-11-00667-f003:**
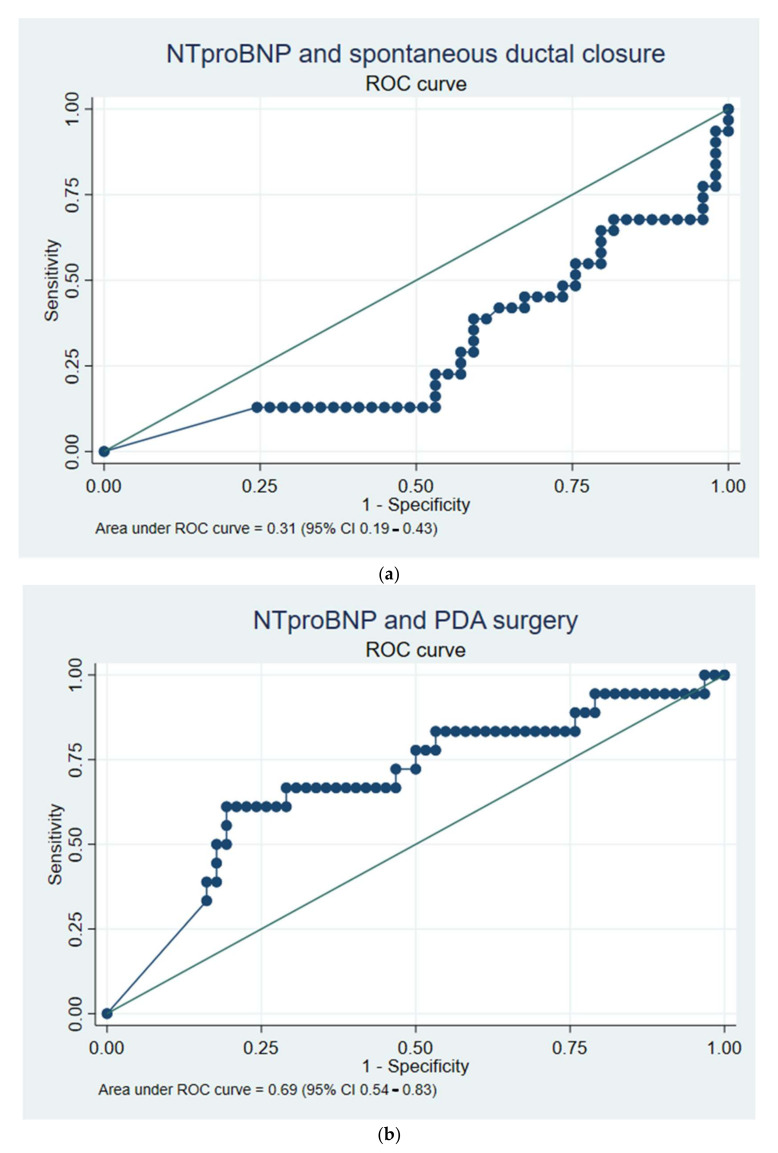
(**a**) The ROC curve is shown for the NTproBNP value on day three in predicting later spontaneous ductal closure in extremely preterm infants (*N* = 80). Abbreviations: ROC: receiver operating characteristics; NTproBNP: N-terminal pro B-type natriuretic peptide. (**b**) The ROC curve is shown for the NTproBNP value on day three in predicting later PDA surgery in extremely preterm infants (*N* = 80). Abbreviations: PDA: patent ductus arteriosus; NTproBNP: N-terminal pro B-type natriuretic peptide; ROC: receiver operating characteristics.

**Table 1 jcm-11-00667-t001:** Characteristics and neonatal outcomes of the infants by PDA treatment.

	No Treatment	Only Ibuprofen Treatment	All Surgery	*p*-Value ^a^
*N* = 98	*n* = 40	*n* = 36	*n* = 22	
Any antenatal steroids, *n* (%)	39 (98)	36 (100)	22 (100)	0.48
Multiple birth, *n* (%)	10 (25)	17 (47)	8 (36)	0.41
Vaginal delivery, *n* (%)	10 (25)	11 (31)	11 (50)	0.06
Chorioamnionitis, *n* (%)	17 (43)	18 (50)	7 (32)	0.62
Gestational age, weeks, mean (SD)	26.4 (1.1)	25.7 (1.2)	24.7 (1.0)	<0.001 *
Gestational age < 25 weeks, *n* (%)	4 (10)	10 (28)	10 (45) ^b^	<0.001 *
Birth weight, grams, mean (SD)	816 (173)	806 (206)	698 (121)	0.03 *
SDS birth weight, mean (SD)	−1.4 (1.4)	−0.9 (1.1)	−0.8 (0.8)	0.11
Female sex, *n* (%)	21 (53)	19 (53)	9 (41)	0.55
Apgar < 7 at 5 min, *n* (%), *N* = 97	15 (38)	18 (51)	13 (59)	0.31
Surfactant within 2 h after birth, *n* (%)	16 (38)	24 (67)	19 (86)	0.001 *
IPPV at start of ibuprofen PDA trx, *n* (%) *N* = 53	NA	12 (33)	10 (45)	NA
Inotropy treatment ≤ 7 days of life ^c^, *n* (%)	4 (10)	6 (17)	0 (0)	0.15
NO treatment ≤ 7 days of life ^c^, *n* (%)	2 (5)	3 (9)	0 (0)	0.36
Postnatal steroid trx ≤ 7 days of life ^d^, *n* (%)	1 (2)	1 (3)	0 (0)	0.99
IVH grade 3 or higher ^e^, *n* (%)	6 (15)	1 (3)	7 (32)	0.008
NEC stage IIb ^f^ or higher, *n* (%)	8 (20)	3 (9)	6 (27)	0.15
Severe BPD ^g^, *n* (%) *N* = 86	4 (10)	1 (3)	7 (32)	0.006
Any postnatal steroids ^h^, *n* (%)	7 (18)	6 (17)	7 (32)	0.34
Death before ductal closure, *n* (%)	6 (15)	5 (14)	0 (0)	NA ^i^
All deaths, *n* (%)	7 (18)	5 (14)	0 (0)	NA ^i^

^a^ Kruskal–Wallis test, Chi-Square test or Fischer´s Exact Probability test used for comparison of the three groups, no treatment, ibuprofen treatment only and all surgery. ^b^ Of the five infants, who received primary PDA surgery, four (80%) were born at <25 weeks gestational age. ^c^ Four infants treated with early inotropy, thereof one also received early NO, died. All the infants receiving NO also received inotropy. ^d^ intranvenous hydrocortisone given^e^ IVH as graded by Papile [24]. ^e,f^ NEC as graded by Bell’s [25]. ^g^ Severe BPD defined as need of ≥30% oxygen or positive pressure ventilation at 36 weeks postmenstrual age in the surviving infants. ^h^ Either intravenous hydrocortisone or betamethasone. ^i^ Not applicable as the infant had to survive until the operation day and in the group of infants who underwent surgery, none died. * Significant differences in overall comparison. Abbreviations: PDA: patent ductus arteriosus; SD: standard deviation; IQR: interquartile range; IPPV: invasive positive pressure ventilation; NA: not applicable; NO: nitric oxide; trx: treatment; IVH: intraventricular haemorrhage; NEC: necrotising enterocolitis; BPD: bronchopulmonary dysplasia.

**Table 2 jcm-11-00667-t002:** Echocardiographic markers of a haemodynamic significant PDA shunt on day three and seven, categorised in: no treatment, only ibuprofen PDA treatment and all surgical PDA treatment.

	No Treatment	Only Ibuprofen Treatment	All Surgery	
	*n* = 40	*n* = 36	*n* = 22	
Day 3 (range 2–4) ^a^				*p*-value ^b^
Bidirectional shunt, or right to left through the duct, *n* (%)	*n* = 26	*n* = 21	*n* = 16	
	8 (31)	1 (5)	3 (19)	0.08
Ductal diameter, mm, median (IQR)	*n* = 22	*n* = 23	*n* = 15	
	1.3 (1.2–1.7)	1.9 (1.7–2.0)	1.7 (1.6–1.9)	0.002 *^†^
Vmax through the duct, m/s, median (IQR)	*n* = 16	*n* = 20	*n* = 12	
	1.6 (1.2–2.4)	1.3 (1.1–1.9)	1.3 (1.2–1.6)	0.33
LA:Ao ratio, median (IQR)	*n* = 27	*n* = 18	*n* = 16	
	1.4 (1.2–1.6)	1.6 (1.4–1.9)	1.7 (1.6–1.8)	<0.001 *^†^
LPA or RPA end-diastolic flow > 0.2 m/s, *n* (%)	*n* = 26	*n* = 18	*n* = 16	
	4 (15)	4 (22)	11 (69)	0.001 *^†^
Signs of ductal steal, *n* (%)	*n* = 31	*n* = 21	*n* = 16	
	2 (7)	8 (38)	7 (44)	0.005 *^†^
Day 7 (range 5–9) ^c,d^				*p*-value ^b^
Bidirectional, or right to left flow through the duct, *n* (%)	*n* = 29	*n* = 33	*n* = 20	
	6 (21)	4 (12)	2 (10)	0.506
Ductal diameter, mm, median (IQR)	*n* = 25	*n* = 31	*n* = 17	
	1.6 (1.4–1.9)	1.6 (1.4–2.0)	1.8 (1.5–2.0)	0.411
Vmax through the duct, m/s, median (IQR)	*n* = 19	*n* = 29	*n* = 17	
	1.8 (1.5–2.7)	1.7 (1.5–2.2)	1.3 (0.9–1.6)	<0.001 *^††^
LA:Ao ratio, median (IQR)	*n* = 31	*n* = 29	*n* = 16	
	1.5 (1.3–1.5)	1.5 (1.4–1.8)	1.6 (1.4–1.7)	0.037 *^††^
LPA or RPA end-diastolic flow > 0.2 m/s, *n* (%)	*n* = 27	*n* = 28	*n* = 15	
	8 (30)	11 (39)	0 (0.0)	0.021 *^††^
Signs of ductal steal, *n* (%)	*n* = 32	*n* = 30	*n* = 19	
	7 (22)	15 (50)	6 (32)	0.064

^a^ Five infants had a closed duct on the day three echocardiography. ^b^ Overall comparison between treatment categories tested with the Kruskal–Wallis test. ^c^ Four infants had a closed duct on the day seven echocardiography, in addition to the five on day three. ^d^ Twenty-eight ínfants had already started ibuprofen treatment. * Significant differences in overall comparison. ^†^ On day 3: In pairwise comparison (Dunn’s test and Chi2 as appropriate with *p* < 0.05), significant differences were detected for ductal diameter and LA:Ao ratio between no treatment and both PDA treatment categories. Significant differences in signs of steal were also detected between no treatment and both PDA treatment categories. Furthermore, significant differences in excessive pulmonary flow were seen when comparing all surgery category with no treatment and ibuprofen treatment only, respectively. ^††^ On day 7: In pairwise comparison significant differences were detected between no PDA treatment and all PDA surgery as well as between all PDA surgery and only ibuprofen treatment. Abbreviations: PDA: patent ductus arteriosus; IQR: interquartile range; LA:Ao: left atrium to aortic root ratio; LPA: left pulmonary artery; RPA: right pulmonary artery; Vmax: maximum velocity.

**Table 3 jcm-11-00667-t003:** NTproBNP levels by PDA treatment category on days three and seven.

PDA Treatment Categories	NTproBNP Day 3 (ng/L) ^a^ Median (IQR), *N* = 85	NTproBNP Day 7 (ng/L) ^a^ Median (IQR), *N* = 95
No treatment, spontaneous PDA closure ≤ 7 days of life	*n* = 5 GA ^b^ = 26.6 (25.9–26.7) ^†^	*n* = 6 GA ^b^ = 26.7 (25.7–27.0) ^†^
	1810 (1760–6000)	1915 (1115–2360)
No treatment, spontaneous PDA closure > 7 days of life	*n* = 32 GA ^b^ = 26.6 (25.4–27.3) ^†^	*n* = 36 GA ^b^ = 26.7 (25.5–27.4) ^†^
	10,900 (6120–19,200)	3735 (1820–12,995)
Only ibuprofen	*n* = 33 GA ^b^ = 25.9 (24.9–26.6) ^†^	*n* = 33 GA ^b^ = 25.9 (24.9–26.6) ^†^
	14,600 (7740–28,100)	5790 (4030–10,400)
All surgery	*n* = 15 GA ^b^ = 25.0 (23.8–25.8) ^†^	*n* = 20 GA ^b^ = 25.0 (23.7–25.5) ^†^
	32,300 (29,100–35,000)	8790 (4810–16,050)
*p*-value ^c^	0.001 *	0.003 **

^a^ Time intervals were 2–4 days for day three and 5–9 days for day seven. All infants with an available value on day three survived. PDA treatment was started on day 7 or earlier in 28/98 infants. ^b^ Gestational age of the infants in each category is shown with median and IQR. ^c^ Overall comparison of the median NTproBNP level between treatment categories tested with Kruskal–Wallis test. * In pairwise comparisons (Dunn’s test): *p* < 0.05 between spontaneous PDA closure ≤ 7days and all other PDA categories, and between all other PDA categories and all surgery. There was no significant difference between no treatment (with open duct beyond first week of life) and ibuprofen treatment only. ** *p* < 0.05 for all pairwise comparisons, except for the difference between ibuprofen only and all surgery which was non-significant. ^†^ In overall comparison there were significant differences in the infants’ median gestational age at birth between the no PDA treatment (both groups combined), only ibuprofen and all surgery group (*p* < 0.001). Abbreviations: NTproBNP: N-terminal pro B-type natriuretic peptide; PDA: patent ductus arteriosus; IQR: interquartile range; GA: gestational age.

**Table 4 jcm-11-00667-t004:** cTnT levels by PDA treatment category on day three and seven days.

PDA Treatment Categories	cTnT Day 3 (ng/L) ^a^ Median (IQR), *N* = 83	cTnT Day 7 (ng/L) ^a^ Median (IQR), *N* = 93
No treatment, spontaneous PDA closure ≤ 7 days of life	*n* = 5	*n* = 4
	171 (105–191)	136 (97–180)
No treatment, spontaneous closure > 7 days of life	*n* = 31	*n* = 36
	172 (105–261)	98 (74–142)
Only ibuprofen	*n* = 33	*n* = 33
	151 (112–213)	96 (86–122)
All surgery	*n* = 14	*n* = 20
	256 (151–317)	130 (96–151)
*p*-value ^b^	0.19	0.26

^a^ The time intervals were 2–4 days for day three and 5–9 days for day seven. PDA treatment was started on day seven or earlier in 28/98 infants. ^b^ Overall comparison between treatment categories tested with Kruskal–Wallis test. Abbreviations: cTnT: cardiac Troponin T; PDA: patent ductus arteriosus; IQR: interquartile range.

## Data Availability

The data presented in this study are available on request from the corresponding author. The data are not publicly available due to ethical concerns.

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
