# Peer review of "Early N-Terminal Pro B-Type Natriuretic Peptide (NTproBNP) Plasma Values and Associations with Patent Ductus Arteriosus Closure and Treatment—An Echocardiography Study of Extremely Preterm Infants"

_jcm, 2022, doi:10.3390/jcm11030667_

Round 1

Reviewer 1 Report

This prospective observational study of preterm infants born at £28 weeks gestational age aim to look at the relationship between spontaneous ductal closure, ductal treatment, gestational age, echocardiographic parameters and biochemical markers (NTproBNP & cTnT) levels.

Hypothesis: Early biochemical markers can identify babies who would receive treatment in this cohort.

This is a carefully followed up cohort with regular echocardiography. While both the ductal outcomes as well as biochemical markers assessment are not novel, this study adds to the evidence base of evolution of ductus in extreme preterm infants.

Introduction: This could be more succinct focussing on studies describing spontaneous closure and/ or use of biochemical markers.

Methods: The methods are described in appropriate details. My specific comments would be:

  1. The treatment protocol is not described. The authors have given reference no 23 as guideline. This is not a published paper. They should consider removing this from paper and briefly summarise their treatment approach or include this guideline in appendix.
  2. Fig 1 – Reasons for non-inclusion in 31 infants
  3. Detailed echocardiographic assessment is described and it appears appropriate. It would strengthen the findings if they have some assessment of inter and intra-observer reproducibility of echocardiographic parameters

Statistical analysis is appropriate for this study.

Results:

The readability of this section could be vastly improved by removing duplicate numbers that are already provided in the tables.

The NTproBNP levels on day 3 showed a poor sensitivity (61%) and specificity(20%) to be useful as a predictive tool for spontaneous closure of ductus.

There are too many Kaplan Meir survival curves given. I think this detracts from the key messages. I would keep the Fig 2a and rest can be moved to supplementary files.

The authors have not described if the addition of biochemical markers over and above the GA and echocardiographic parameters add to the predictive power. Even if the combination of factors do not improve the ability to predict, this should be reported.

Discussion:

Line 334:

“In this observational cohort study of extremely preterm infants, the proportion of an echocardiographically verified ductal closure was high”

The reported rates of spontaneous closure are similar to what has been reported by de Klerk in his systemic review (34% closure by Day 4 and 41% by Day 7) and study by Pereira et al.  2018* who reported a closure rate of 32% by Day 3).

*Pereira et al. Early echocardiography does not predict subsequent treatment of symptomatic Patent Ductus Arteriosus in extremely preterm infant. Acta Paediatrica 2018, 11:107;1909-16

The discussion is too broad. It should focus on the findings of this paper and how it compares with previously reported study and implications of this study findings.

Conclusion:

The authors state, Line 445-447

The combination of echocardiographic markers for ductal severity and NTproBNP measured during the first postnatal days is helpful in identifying infants at risk of developing a haemodynamic significant PDA in need of treatment.

I could not see any evidence to support this statement in the paper (Please see the last paragraph in the results section).

Author Response

Please see the attached pdf file/thank you

Reviewer 2 Report

Gudmundsdottir A et al. described their PDA-related data in relationship to troponin and NTproBNP in “Early N-Terminal pro B-Type natriuretic peptide plasma values associated with spontaneous patent ductus arteriosus closure rate…”. This is a prospective study that represents a lot of work and a breadth of information that, although observational, is valuable to the readership. They found a relationship between NTproBNP and type of ductal management strategies in their unit (which tends to be guided by the echocardiography profile). Here are some suggestions regarding this manuscript to take into consideration

  1. I could not access the supplementary files sadly.
  2. The title is misleading. It pretends that it looks at a population of patients treated conservatively and allowed to spontaneously close their ductus like PUBMED: 34815521, 28701390, 25169243, 30759169. However, there is 98 infants included in the study, out of which ALL the surviving neonates <25 weeks were exposed to a strategy (would not call it “a treatment”) to accelerate ductal closure (such as: NSAIDs or ligation). Out of the 40 infants on 98 not exposed to such a strategy, 4 were <25 weeks and died before exposure (page 6). Out of the 36 of 40 left, there were 8 remaining infants that passed away (which may have ended up ligated or exposed to NSAIDs in their center) – so really it leaves us with 28 infants with a permissive approach allowing for spontaneous closure to occur (out of which 5 were transferred out without evaluation for closure)… I would adjust the title to really reflect the message of the article.
  3. Were the data extractors from the echocardiography masked to the results of the NTproBNP and Troponin values?
  4. The authors should consider correlating the NTproBNP with LA/Ao measurements, since they postulate that the dilation of LA/LV cavity is at the root source of this rise in NTproBNP. It could be a clean correlation analysis provided as supplementary material. The correlation needs to take into consideration the repeated measure model – such as with a random mixed effect model with random slope and intercept.
  5. I would avoid using the terminology: “need pharmacological PDA treatment”. There is a body of literature outlining the controversy about the “need” to expose preterm newborns to any strategies to accelerate ductal closure. As such, I would stay factual: “infants that were exposed to pharmacological strategies to accelerate ductal closure, such as with ibuprofen”. Literature to consider exposing or putting in relationship of results: PUBMED: 34815521, 28701390, 25169243, 30759169, 28724506.
  6. Any use of paracetamol/acetaminophen in their cohort?
  7. Any exposure to post-natal steroids in these infants? There is some data outlining the impact of dexamethasone on ductal closure (PMID: 9568218). Steroids may also impact cardiac chambers and lead to a rise in NT-proBNP (through various mechanisms such as the rise in systemic afterload).
  8. Authors should consider presenting their rate of BPD, NEC and IVH grade 3 or more since they mention that these are outcomes thought to be influenced highly by the ductal management. It could be added in Table 1. Now, knowing that BPD is highly associated with GA, it is likely that the NTproBNP will be high in those with more BPD.
  9. Line 202 of page 6 should be “country” instead of “county”
  10. The authors present “death before ductal closure” but should consider presenting “all deaths” in the table 1.
  11. I am surprised on the high rate of exposure to NSAIDs or ligation in this population of newborn. 100% of surviving <25 weeks; 58% of >=25 weeks surviving (38/66 from what I can calculate based on their flow diagram). I see that this data comes from a population born between 2012 and 2014 – did their practice chance since that time? Are they less exposing their preterm to strategies accelerating ductal closure?
  12. Any babies on inotropes at the time of the echocardiography / markers sampling. Inotopes may affect cardiac function and afterload and may also be reflected in a rise in NTproBNP and troponin.
  13. It would have been interesting to have values of systemic blood pressure at the time of the echocardiography, as this may also impact (as a reflection of afterload or diastolic steal), the values of NTproBNP
  14. Many infants had a bidirectional / right to left flow at ductal level at day 7 of life. Many of these infants were exposed to ductal closure strategies, knowing that there may be underlying high PVR. What is the impact of high PVR on secretions of NTproBNP? Knowing that it may also impact RV afterload? Also, what was the rate of small for gestational age in their population?
  15. The babies exposed to no-treatment were often those that were more mature. Hence they have a tendency to close earlier. Could NTproBNP only be a reflection of gestational age changes? Should all the associations of Table 3a be adjusted for GA at birth?
  16. I have some concerns about putting GA in a ROC model since there is a large part of confounding by indication. All their <25 weeks were treated (except the 4 that died). This part should be probably considered for removal.
  17. In the discussion: “predicted both early spontaneous closure and risk of later PDA surgery”. To be factual: “predicted both early spontaneous closure and exposure to PDA ligation in our cohort”.
  18. When discussing – “the rate of spontaneous closure in infants born before 25 weeks has been difficult to establish.” - Paper (PMID: 34815521) has outlined that many infants are able to close their ductus even between 23 to 25 weeks GA at birth. One can wonder what would be the advantage of measuring NTproBNP if we do not need to attempt accelerated closure? It would have been interesting to see if the NTproBNP may be associated with other meaningful outcomes in their cohort – BPD, IVHG3 or more, NEC, death, length of hospitalization. I think the literature is now going beyond the prediction of closure vs non closure.
  19. The authors mention: “it seems plausible to regard the infants in this study that later had PDA surgery, as representatives of infants with the grade of ductal severity that warrants special attention”. Why is that? Could the author provide a quantifiable metric to justify this statement. Decision to expose to ligation or not is often arbitrary. The Table 2 outlines that Ductal Diameter, Vmax, LA/Ao, Signs of ductal steal are relatively similar between the “Only Ibu” and the “PDA” (actually even better at Day 7).
  20. My understanding is that the sampling of blood is 0.5 mL for NTproBNP. As such, it is not a huge amount of blood. What is the cost of such analysis at the scale of all these newborns? Who should be the newborns in which we should be measuring NTproBNP?

Author Response

please see pdf file, thank you

Round 2

Reviewer 2 Report

The authors have responded appropriately to all my comments and questions, and have made the appropriate adjustments. Here are only minor elements:

  • Table 1 legend: Hydrocortison should be hydrocortisone
  • Table 1 legend: the reference for Volpe and Bell are not provided
  • The sentence: “The study investigator with extensive training in echocardiography (AG) re-examined all the echocardiographies according to a predefined protocol (designed by L-AM and AG) and these were the measurements accepted for the analyses”. Were these the ECHO measurements that were used for the final analysis and, if so, was AG blinded to the values of the biomarkers at the time of the ECHO measurements. That should be mentioned in their methods. If AG was blinded – this is a strength to the study results. If not blinded (i.e. AG knew the biomarkers value for each ECHO during data extraction of the ECHO), it is a limitation.

Author Response

Thank you for your excellent comments.

We have revised the manuscript according to the comments.

  • Table 1 legend: Hydrocortison should be hydrocortisone: THIS HAS BEEN CHANGED
  • Table 1 legend: the reference for Volpe and Bell are not provided: THE REFERENCES HAVE BEEN PROVIDED, NR 24 AND 25. IT WAS PAPILE AND NOT VOLPE, WE APOLOGISE FOR THAT.
  • The sentence: “The study investigator with extensive training in echocardiography (AG) re-examined all the echocardiographies according to a predefined protocol (designed by L-AM and AG) and these were the measurements accepted for the analyses”. Were these the ECHO measurements that were used for the final analysis and, if so, was AG blinded to the values of the biomarkers at the time of the ECHO measurements. That should be mentioned in their methods. If AG was blinded – this is a strength to the study results. If not blinded (i.e. AG knew the biomarkers value for each ECHO during data extraction of the ECHO), it is a limitation. THANK YOU FOR POINTING THIS IMPORTANT POINT, THE REVIEWER, AG WAS BLINDED TO THE BIOMARKER RESULTS WHEN REVIEWING THE ECHOCARDIOGRAPHIES, AS ANOTHER CO-AUTHOR COLLECTED THE BIOMARKER DATA (O.PICARD). WE HAVE CHANGED THE TEXT IN THE METHODS SECTIONS, LINES 109-110 AND IN THE DISCUSSION, LINES 489-90, TO HIGHLIGHT THIS.

Additionally, we have revised typos in the manuscript and graphical abstract.  Furthermore, in the results section a reference in text to supplemental figure 3 and 4 was both missing (supplemental figure 4) and mixed up (4 was supposed to be 3 etc). This has been revised. Supplemental figure 3 (earlier 4), the graph titles have been revised to be in concordance with the manuscript. (strategy surgery->PDA surgery; and a note that LA:Ao ratio and ductal diameter were on day three).

Best regards

Anna Gudmundsdottir